# Thiophene-Based Trimers and Their Bioapplications: An Overview

**DOI:** 10.3390/polym13121977

**Published:** 2021-06-16

**Authors:** Lorenzo Vallan, Emin Istif, I. Jénnifer Gómez, Nuria Alegret, Daniele Mantione

**Affiliations:** 1Laboratoire de Chimie des Polymères Organiques (LCPO—UMR 5629), Université de Bordeaux, Bordeaux INP, CNRS F, 33607 Pessac, France; Lorenzo.Vallan@enscbp.fr; 2Department of Mechanical Engineering, Koç University, Rumelifeneri Yolu, Sarıyer, Istanbul 34450, Turkey; eistif@ku.edu.tr; 3Department of Condensed Matter Physics, Faculty of Science, Masaryk University, 61137 Brno, Czech Republic; gomez.perez@ceitec.muni.cz; 4POLYMAT and Departamento de Química Aplicada, University of the Basque Country, UPV/EHU, 20018 Donostia-San Sebastián, Spain

**Keywords:** conjugated polymers, polythiophenes, terthiophenes, thiophene trimers, biosensing, photosensitizers

## Abstract

Certainly, the success of polythiophenes is due in the first place to their outstanding electronic properties and superior processability. Nevertheless, there are additional reasons that contribute to arouse the scientific interest around these materials. Among these, the large variety of chemical modifications that is possible to perform on the thiophene ring is a precious aspect. In particular, a turning point was marked by the diffusion of synthetic strategies for the preparation of terthiophenes: the vast richness of approaches today available for the easy customization of these structures allows the finetuning of their chemical, physical, and optical properties. Therefore, terthiophene derivatives have become an extremely versatile class of compounds both for direct application or for the preparation of electronic functional polymers. Moreover, their biocompatibility and ease of functionalization make them appealing for biology and medical research, as it testifies to the blossoming of studies in these fields in which they are involved. It is thus with the willingness to guide the reader through all the possibilities offered by these structures that this review elucidates the synthetic methods and describes the full chemical variety of terthiophenes and their derivatives. In the final part, an in-depth presentation of their numerous bioapplications intends to provide a complete picture of the state of the art.

## 1. Introduction

Today, conductive polymers have become inalienable components of a wide spectrum of advanced technologies, including sensor and biosensors, batteries, solar cells, LEDs, and organic transistors [1,2,3,4,5,6]. Among conductive polymers, poly(3,4-ethylenedioxothiophene) (PEDOT) is undoubtedly the most widespread. In fact, a set of useful properties, such as high conductivity, superior photostability, low redox potential, transparency, and good processability, contributes to the exceptionality of this polymer [7]. Typically purchased as a water dispersion of PEDOT:PSS (poly(styrene)sulfonate), PEDOT is cheap and readily available for research purposes. As an alternative, PEDOT can be easily prepared by polymerization from the EDOT monomer, allowing the preparation of films on a wide range of substrates. While for the above-mentioned reasons PEDOT was quickly establishing itself as the gold standard of conductive polymers, the progress in organic synthesis gradually disclosed new possibilities for the easier preparation and modification of alternative thiophene-based structures, including thiophene and EDOT monomers carrying different functional groups. Therefore, a new research branch focused on the preparation of more complex thiophene-based structures has blossomed, taking advantage of the introduction of functionalities for the tuning of physical, chemical, and optical properties towards selected and specific applications. Nevertheless, the use of these functional structures is still relatively narrow with respect to the ubiquity of PEDOT. The main reason behind this are probably the costs. The synthesis of functionalized monomers requires organic laboratory facilities, specific knowledge, and time. On the other hand, dozens of publications per year demonstrate that the modification of thiophene-based monomers is a tremendous approach for the preparation of advanced functional materials.

In this framework, the preparation of thiophene-based trimers has proven to be an interesting solution, resulting in a relatively simple synthesis and with the capability to afford complex structures with variable functional groups. In fact, the compatibility with the polymerization conditions of the monomer side groups connected to position 3–4 or 2–3 of the thiophene or the EDOT, respectively, (see Scheme 1) is limited. For example, the common precursor hydroxymethyl-EDOT cannot be converted to carboxylic acid without triggering the undesired polymerization. In some cases, the functional group contributes to lowering the oxidation potential of the monomer, making it particularly sensitive towards uncontrolled polymerization, while, in other cases, the influence of the side group has the opposite effect, hampering the polymerization [8]. Many of these problems can be overcome by the synthesis of terthiophene (thiophene trimer): the presence of a side group in position 3′ or 4′ on the central unit will interfere less with the trimer polymerization in position 5 and 5″ (Scheme 1), thus allowing the introduction on the polymer of functional groups otherwise incompatible. Moreover, in contrast to monomers, trimers allow us to obtain perfectly alternated copolymers. This opportunity is undeniably useful when it is pursued with the preparation of donor–acceptor copolymers with tunable bandgaps. Additionally, being symmetrical molecules, polymerizable trimers are frequently easier to prepare with respect to dimers, especially when very efficient synthetic strategies, such as Suzuki coupling, are covered. Furthermore, although trimers are generally synthesized as polymers precursors, they have also been exploited for several direct applications, thanks to their interesting optical properties that result from the large electron delocalization. Finally, both trimers and polymers have been demonstrated to be safe and biocompatible materials.

Terthiophenes can therefore be an elegant and efficient solution for those who may look for novel optoelectronic materials and photoactive molecules for bioapplications. With that in mind, this review aimed to offer an exhaustive walkthrough over the thiophene-based trimer structures reported in the literature until now. Thus, the first part is dedicated to the presentation of the different synthetic strategies employed for the trimer scaffold synthesis, independently from the functional side groups. The central part is a compendium of all the functional groups that have been introduced on thiophene-based trimers, classified per elementary organic group; in order to skim the literature, polymerizable thiophene-trimer was preferred (i.e., bearing hydrogen atoms in positions 5 and 5″). Finally, the latter part is focused on the most promising applications of trimer-based materials in biology and medical science. In particular, sensing, photodynamic therapy, drug delivery, tissue engineering, antibacterial, and pesticide activity are discussed.

## 2. Synthetic Methodologies

The first isolated terthiophene was reported in 1942 andwas obtained as a byproduct in the synthesis of bisthiophene [9]. The Ullmann coupling method was exploited for the bisthiophene synthesis by treating 2-iodothiophene with copper. Even if the main product of the reaction was bithiophene, terthiophene was also isolated in a small amount [10]. It is worth mention that, in 1947, terthiophene was extracted from a natural source: marigold flowers [11]. Since then, various approaches have been developed for the synthesis of terthiophenes, which can be grouped into two routes: the C–C bond formation between thiophenes or the ring closure reaction from precursor molecules (Scheme 2). 

### 2.1. Carbon–Carbon (C–C) Cross-Coupling Methods

Metal-promoted C–C coupling reactions are widely reported in the literature because of the ease of the experimental protocols and the high yield related. They consist in the formation of a new C–C bond with the aid of a metal catalyst. A description of the different C–C cross-coupling reactions is presented in the following sections. 

#### 2.1.1. Kumada Reaction 

Since the moment that conducting polymers were discovered, immediate attention was given to the synthesis of polyaromatic polymers such as polythiophenes [12]. In 1980, the Kumada reaction [13] was used for the synthesis of conducting polymers by Grignard cross–coupling of heterocyclic compounds [14,15]. Afterwards, the Kumada reaction was reported several times as an efficient method for the synthesis of various types of oligothiophenes. The Grignard reagent preparation and the further cross-coupling are usually performed in a one-pot reaction. First, 2-bromothiophene in ether-based solvent reacts with Mg in order to generate the corresponding organometallic compound; second, the addition of dibromothiophene derivatives and a metal catalyst (typically nickel or palladium complexes, 0.01–0.1 eq.) allows the elimination of magnesium bromide and the formation of a C–C bond. Organolithium intermediates could be useful to prepare derivatives bearing a bromine and a magnesium bromide on the same ring. The reaction proceeds under mild conditions with high reaction yields. A representative reaction scheme for terthiophene synthesis by Kumada coupling is depicted in Scheme 3. Terthiophenes synthesized by Kumada coupling have been used for direct applications or as intermediates for further reactions [16,17,18,19,20,21,22,23,24]. 

The Kumada reaction has some drawbacks with respect to other types of metal-catalyzed cross-coupling reactions. In fact, Mg or Li reagents are also reactive towards some types of organic groups, including aldehydes and ketones, affording undesired byproducts when they are present as thiophene side groups [10]. To overcome this limitation, protecting groups can be used to mask the sensitive functional groups before the Kumada reaction. Another drawback is the high sensitivity to water and air of this reaction, which requires dry and oxygen-free reagents and solvents to achieve good yields. 

#### 2.1.2. Stille Reaction 

Stille cross-coupling consists of the palladium-catalyzed C–C bond formation between an organotin aromatic compound and an aryl halide. Several types of palladium complex catalysts have been developed, such as Pd(PPh_3_)_4_, Pd(PPh_3_)_2_Cl_2_, Pd(dppp)Cl_2_, Pd(dppf)Cl_2_, and Pd_2_(dba)_3_. This reaction is one of the commonest methods for the synthesis of terthiophene and its derivatives, since it requires mild reaction conditions, is regioselective, and is compatible with many functional groups, including aldehydes, ketones, alcohols, nitriles, and esters [10]. The synthesis of unsubstituted terthiophene by Stille coupling is performed using 2,5-dibromo- or 2,5-diiodothiophene with organotin derivatives (such as tributhyl(2-thienyl)tin) in presence of the Pd catalyst [25,26,27,28,29]. The reaction is illustrated in Scheme 4. An alternative approach was covered in the work of Kamal et al., where the Stille reaction was performed between a bithiophene-based organotin compound and a 2-bromothiophene. Although for the synthesis of terthiophenes by Stille coupling can afford high conversion yields (around 75–90%), the acute toxicity of organotin compounds should be of concern to those who may perform this reaction.

#### 2.1.3. Suzuki Reaction

In the synthesis of terthiophenes by Suzuki cross-coupling, a palladium complex catalyzes the C–C bond formation between a 2,5-dibromo- or 2,5-diiodothiophene and a thiophene boronic acid or boronate ester (pinacol ester) (Scheme 5). The Suzuki reaction is today the most common choice for the synthesis of terthiophene derivatives, thanks to its good yield (50 to 90%) and to its compatibility with the presence of functional groups both on the thiophene halide and on the thiophene boronate [30,31,32]. After the discovery of this reaction in 1979, many studies assessed its ability to work for different heterocyclic compounds. One of the first applications of Suzuki coupling on the synthesis of terthiophenes was presented by Gronowitz et al. [33], which reported the coupling of 2-thiophene boronic acid with 2,5-dibromothiophene with a 40% yield. Interestingly, they noticed that using an excess of 20% of boronic acid in the coupling reaction prevents the formation of mono-coupled byproducts, which are difficult to separate from the product. In the case of electron-rich heteroaromatics, the unwanted deboronation of organoboron compounds may also occur [33]. Melucci et al. demonstrated a facile, solvent-free and microwave-assisted approach for synthesis of various oligothiophenes derivatives via the Suzuki reaction. They reported the coupling of dibromothiophene with thiophene boronic acid using various Pd catalysts (5% mol with respect to boronic acid) in order to optimize the reaction yield [34]. Another microwave-based trimer synthesis was developed by Alesi et al. for the preparation of highly pure thiophene oligomers using silica- and chitosan-supported Pd complexes. The trimer has also been synthesized using 2,5-diiodothiophene instead of 2,5-dibromothiophene [35]. A similar approach was covered by Di Maria et al., using chitosan-supported Pd for the synthesis of terthiophene for cell imaging applications [36]. 

### 2.2. Ring Closure Reactions 

As an alternative to the C–C cross-coupling methods, ring closure reactions can be exploited for the terthiophene formation. Here, the central thiophene ring is formed last by a cyclization step on a precursor made of two thiophene rings connected by different possible structures.

#### 2.2.1. Cyclization of 1,3-diynes

This strategy involves the formation of the central thiophene ring from a bis-thienylbutadiyne. This step gives good yields, but requires the synthesis of the diyne precursor. The preparation of diynes can be performed by oxidative coupling of 2-thienylacetylenes and further cyclization of the diyne group (Scheme 6) [37]. 

Beny et al. first developed this strategy for the synthesis of terthiophene [37]. 2-thienylacetylenes were synthesized in two steps from 2-thiophenealdehyde: first, this molecule was converted to 2-(2,2-dibromoethenyl)thiophene using carbon tetrabro-mide (Corey-Fuchs procedure), then the treatment with n-butyllithium caused the de-hydrobromination and the formation of the alkyne. The oxidative coupling of 2-thienylacetylenes was performed with copper chloride. In the last step, a [4+2] peri-cyclic reaction of the 1,3-diyne group is driven by sodium sulfide, affording the final terthiophene structure in a total yield of 46%. Later, Perrine et al. increased the overall reaction yield of the terthiophene synthesis to 77% by improving the purification steps [38]. In another study, Carpita et al. used this synthetic procedure for the synthesis of terthiophene and thiophene-furan-thiophene type oligomers as antifungal agents [39]. This method was also applied for preparing terthiophene as intermediate for further modification, i.e. to obtain diiodoterthiophene [40], and thiolate-terthiophene [41]. In the last decade, many studies focused on the improvement and readaptation of this cyclization reaction. Zheng et al. described a one-pot, two steps radical reaction for cy-clization of diacetlylene precursor in the presence of KOH and sodium disulfide in DMSO. The reaction yielded the terthiophene without the use of a transition metal catalyst [42]. In another study, the synthesis of terthiophene was achieved from the diacetlylene precursor with elemental sulfur and NaOtBu as reactants [43]. Urselmann et al. reported a one-pot synthesis of terthiophene where 2-iodothiophene is converted to terthiophene using a Pd/Cu-catalyzed Sonogashira–Glaser process followed by sul-fide-mediated cyclization [44].

#### 2.2.2. Cyclization of 1,4-diketone

Another convenient method for the synthesis of terthiophenes by cyclization employs thiophene-substituted 1,4-diketones. Wynberg et al. reported a pathway for the synthesis of 1,4-dithienyl 1,4-diketone starting from 2-acetylthiophene. In here, a Mannich reaction converts 2-acetylthiophene into the corresponding β-aminoketone, which reacts in a further step with an activated thiophene aldehyde (Stetter reaction) to give the 1,4-di-(2′-thienyl)-1,4-butanedione in 70% yield as shown in Scheme 7a [45]. The synthesis of 1,4-dithienyl 1,4-diketones can also be achieved via Friedel-Crafts acylation, i.e. two equivalents of thiophene are made react with succinyl chloride and AlCl3 (Scheme 7b) [46]. An alternative approach consists of the synthesis of 1,4-di-(2′-thienyl)-1,4-butanedione by oxidation of the silyl ether of 2-acetylthiophene [47]. Once the diketone product has been obtained, it can be cyclized into the terthiophene by means of different strategies, including treatment with H2S and HCl (Paal-Knorr synthesis), phosphorous (V) sulfide or Lawesson’s reagent (L.R.) [46,48,49,50,51].

### 2.3. Other Synthetic Strategies

Ben-Haida et al. developed a unique approach, consisting of the cleavage of polymer-supported aryl 2-thienyl ketones using a mixture of potassium tert-butoxide and water. The cleavage reaction affords the terthiophene as product [52]. Leriche et al. reported in their study the formation of terthiophenes as side product: their work pursued the synthesis of star-shaped phosphorous oligothiophene derivatives, however, during the coupling reaction, with a stoichiometric amount of 2-tributylstannylthiophene and in the presence of Pd(PPh3)4 as catalyst undesired terthiophene was obtained in 20% yield [53]. 

## 3. Trimer Structures

### 3.1. Unsubstituted EDOT-Containing Thiophene-Based Trimers

In this class are included all the thiophene-based trimers composed by at least one EDOT unit. The introduction of EDOT in the trimer is a straightforward approach for tuning the electrical and optical properties of the whole structure. Different triads built from different combinations of thiophene and EDOT units have been reported, including EDOT-EDOT-EDOT, EDOT-Thiophene-EDOT and Thiophene-EDOT-Thiophene. Cross-coupling reactions for the C-C bond formation between EDOT and thiophene monomers have been widely exploited for the purpose. 

Grignard/Kumada reaction is one of the most reported synthesis methods for the preparation of unsubstituted EDOT trimers (TerEDOT). In here, an EDOT halide initially reacts with Mg to form the Grignard reagent. Afterwards, dibromo- or diiodoEDOT is added and a Ni(II) catalyst promotes the coupling to give terEDOT [54,55,56,57,58,59]. Kumada reaction was also applied to the synthesis of other unsubstituted EDOT-based trimers, such as thiophene-EDOT-thiophene [60,61] or EDOT-Thiophene-EDOT [62,63]. In alternative, Suzuki [64,65,66] and Stille [67,68,69,70] coupling reactions offer valuable routes for the synthesis of EDOT-based trimers. EDOT organoboron and organotin compounds can be easily synthesized and further coupled with aryl-halide to complete the synthesis of EDOT-based trimers. Lastly, Borghese et al. developed a novel synthetic method for the preparation of EDOT-based trimers consisting of a direct regioselective C–H arylation reaction. The reaction takes place between 2-bromothiophene and EDOT and it is catalyzed by Pd(OAc)_2_ or Pd(Cl)_2_. This approach provides a simple way for the synthesis of oligothiophene series [71]. 

### 3.2. Saturated Aliphatic Substituents

Terthiophenes substituted with one or more bare aliphatic chains are among the most reported types of thiophene-based trimers (examples in Scheme 8). One of the first reports dates back to 1960 and exploit a synthetic pathway discovered in 1947 [72]. It consists of an Ullmann coupling using iodothiophene derivates bearing one or two methyl in position 3 and/or 4 [9,73]. The methyl groups can be also inserted on the central thiophene ring through a [4+2] cycloaddition reaction on di-carbonyl compounds that involves elemental sulfur [37] or the Lawesson’s reagent [74]. Recently di- [75] and monomethyl [16] derivates were prepared through Ni-catalyzed Kumada reactions [17,76,77], as well as ethyl derivatives [78,79]. Monosubstituted trimers have been obtained with a large variety of alkyl groups: by Kumada reaction the octyl [17,77], decyl [80] or dodecyl [81] side groups, by cyclization reaction with substituted diketones and the Lawesson’s reagent in the case of butyl [82], and dodecyl [82], or with phosphorus pentasulfide in the case of eptyl [83] and octadecyl [83]. Pd-catalyzed Suzuki coupling was exploited for the obtaining of octyl derivates [33,84]. The hexyl side chain is among the commonest alkyl moieties in trimer derivatives. Monohexyl trimers have been synthesized by Pd-catalyzed Kumada [85] and Stille coupling [86,87]. Dihexyl trimers substituted in 3 and 3″ were prepared via Ni- [88,89,90,91,92,93,94] or Pd-catalyzed Kumada coupling [95], Suzuki coupling [96,97] and Stille coupling [98]. Dihexyl derivates in position 3′,4′ were obtained through a variant of Ni-catalyzed Kumada reaction [99]. Disubstituted trimers have been achieved also with di-methyl in 4,4″ using Lawesson’s reagent [100] or in 3,3″ using a Pd-Grignard [101] or Pd-halogen coupling [102], di-butyl chains in 3′-4′ either using Ni-Grignard [103,104] or using Pd-Grignard [105,106], di-tert-butyl 3′,4′ using a Zn-Grignard [107], di-octyl in 3,3″ with Ni-Grignard [108,109,110,111,112,113,114], Stille [115,116], Suzuki coupling [117] and halogen substitution [118], di-decyl in 3′,4′ using a Stille coupling [119], di-undecyl [120] dodecyl via Suzuki [121,122] and di-hexadecyl [123] using metal-Grignard reaction pathways.

Additionally, branched aliphatic chains have been introduced on trimers: mono- (3′) and disubstituted (3′-3″) 2-ethylhexyl have been obtained by Ni-catalyzed Kumada [125,127] or Stille coupling [128]. Stille Coupling was also exploited to insert in 3 and 3″ positions two 2,7-dimethyloctyl chains [129]. Lastly, a bis-2,2-dimethyl-butyl group was located in position 3 and 3″ through a variation of the Ni-catalyzed Kumada reaction [130]. Although less common, some trimers substituted with more than two alkyl chains have been reported. For example, Tri- [131] and tetramethyl [124,132] derivates have been synthesize following a Ni/Pd-catalyzed Kumada reaction as well as trioctyl derivates by means of the same procedure [133], by Stille coupling [134] or by Suzuki coupling [135]. Trihexyl substituted trimers were obtained via Kumada coupling [136,137,138], Stille coupling [139], Suzuki coupling [140], Fe-catalyzed Grignard [141] nucleophilic addition [142] and TEMPO-catalyzed Kumada reaction [143,144,145]. More recently microwave-assisted reactions were successfully employed for the introduction of trihexyl [146] and 3,3″-didecyl [126] groups with an adapted Stille coupling and of methyl and hexyl groups by means of decarboxylative Pd-catalyzed cross-coupling [147].

### 3.3. Unsaturated Aliphatic Substituents

Compared to the saturated alkyl derivates, unsaturated derivates are less reported. Trimers with a double-bond in 3′ position were obtained using Wittig reaction [148], metal-catalyzed Grignard reaction [149,150] and Sonogashira coupling [151]. This last strategy was used with a EDOT-thiophene-EDOT trimer [152]. Notably, an unsaturated alkyl trimer was obtained by formation of the central unit with a diketone derivate and P_4_S_10_ [153]. Regarding the alkyne functionality, the favored synthetic strategy for the insertion of triple bonds is the Sonogashira cross-coupling, consisting of the reaction between a bromoalkyl substituted trimer with an alkyne. While gaseous acetylene is normally unusable, trimethylsilyl acetylene or 2-methyl-3-butyn-2-ol can be used for the introduction of a triple bond synthon. A further deprotection step is needed in order to obtain the ethynyl moiety. Through this method, trimers substituted in position 3′ [154,155] and 3′,4′ [156] were obtained. Sonogashira coupling with 2-methyl-3-butyn-2-ol followed by deprotection with potassium hydroxide provided monosubstituted trimers [157,158,159]. Tetrabutylammonium fluoride has been also reported as efficient deprotecting agent [160,161]. Wittig reaction with an halogen derivate has also been used to form an alkene and, subsequently an triple in bond in 3′ by intramolecular elimination [162]. Some examples are shown in Scheme 9.

### 3.4. Nitro Groups

Nitro-substituted thiophene trimers are synthesized by previous nitration of the thiophene monomer, followed by carbon-carbon cross coupling. Nitroderivatives of thiophene trimers are usually employed as intermediates for the synthesis of amino-substituted trimers. Commonly, nitro groups are introduced in positions 3 and 4 of 2,5-dibromothiophene. The resulting 2,5-dibromo-3,4-dinitrothiophene is coupled with organotin or organoboron thiophene derivatives in a metal-catalyzed Stille or Suzuki reaction, thus affording 3′,4′-dinitro trimers [27,163,164,165,166,167,168,169,170,171,172,173,174,175,176,177,178,179]. In some cases, the organotin or organoboron thiophene derivatives carried alkyl or ether chains, that were in this way introduced on the resulting dinitroterthiophene [101,180,181,182,183,184,185,186,187,188,189,190]. In the work of Zotti et al., two EDOT monomers were coupled with the central dinitrothiophene with a yield of 52% [191]. An alternative approach for the synthesis of nitrated thiophene trimers was reported by Leitch et al. [192]. In here, the nitration of the trimer was performed using nitric acid and acetic acid. Although this method conveniently involves only one step starting from commercially available terthiophene, the disadvantage is that it is not regioselective: the direct nitration of terthiophene yields a mixture of four different nitro- regioisomers and column cromatography separation is required. Some examples are shown in Scheme 10.

Trimers carrying nitro groups which are not directly linked to the thiophene ring are also well-documented in literature. For example, nitrophenyl was employed as intermediate groups for obtaining aminophenyl-functionalized trimers [193,194,195]. Overall, when nitro groups were connected directly to the thiophene ring, they were used as intermediates for the synthesis of amine-based trimers. 

### 3.5. Amines 

Amines are conveniently obtained by reduction of nitro groups. Commonly, they are introduced in the positions 3′ and 4′ of the trimer [101,163,165,190,196], but there are examples of trimers carrying amines in 3 and 3″ position [197] and even in 4 and 4″ position [198]. Nitro- or dinitrothiophene is converted to amino- or diaminothiophene by using Sn metal or SnCl_2_ and hydrochloridric acid [101,177,199]. Nevertheless, in order to form the trimer, the amino groups on the monomers should be protected first with tert-Butyloxycarbonyl (Boc), since free amines can interfere in the coupling reaction affording undesired side products [197,200]. For this reason, nitro groups are sometimes reduced after the formation of the trimer. In this case, iron metal in acetic acid solvent has been used as an alternative to SnCl_2_/HCl for the nitro reduction [165,171]. Other reduction methods involve H_2_ on Ni in ethanol [185], H_2_ on Pd/C in ethyl acetate [201] or Zn metal in acetic acid [184]. When directly connected to the thiophene ring, amines are frequently exploited as electron donors. Thiophenes carrying amines can be coupled with nitro- or imido- thiophenes: the presence of both electron donating and electron withdrawing pending groups confers to the resulting trimers a strong zwitterionic-like character [202]. 3,4-diaminothiophene is a widely used trimers building block, not only because of amines electron donating properties, but also as an intermediate for the synthesis of extended conjugated systems such as thienopyrazines [167,199,203], thienothiadiazoles [173,186,190,204], thienoimidazoles [171,205], thienoselenadiazoles [206], and more complex aromatic structures based on those. The preparation of trimers carrying N-linked amide [207], carbamate [197], ethyloxyamyl [208] and oxamate [209] groups is obtained by amine acylation with the corresponding acyl chlorides or anhydrides. Trimers carrying aliphatic and aromatic amines not directly linked to the thiophene ring are also well-documented in literature. For example, amino-styril [210,211,212] and aminopyrimidyl [212,213,214,215,216,217,218] trimers were abundantly employed for sensing and biological applications, as well as aminoacid-linked trimers [200,219]. Some examples are shown in Scheme 11.

### 3.6. Nitriles 

Because of their strong electron withdrawing character, cyano groups (-CN) are introduced on thiophene- or EDOT-based trimers in order to tune their optical and electronic properties. In literature, several studies reports that cyano groups can be incorporated onto different trimer positions by exploiting different synthetic methodologies. However, Suzuki and Stille couling reaction are not widely reported for the synthesis of these trimer derivatives. The principal reasons are two: the difficulties of bromination, especially when the cyano groups are in the 3 position on thiophene and the challenge in the preparation of organotin or organoboron version of a cyano derivate. Grignard reaction is neither a safe route due to high chemical reactivity of the nitrile groups. Common methods for the preparation of cyano trimers are using a palladium-catalyzed coupling reaction via organotin or organozinc intermediates and direct introduction of cyano groups. Organozinc intermediates are formed on the cyano thiophene derivatives, and further the intermediates are coupled with bromo thiophenes in presence of Pd catalysts [220,221,222,223]. Some examples are shown in Scheme 12.

Trimers carrying cyano groups which are not directly linked to the thiophene ring are also well-documented in literature. Using p-toluenesulfonylmethyl isocyanide or Lawesson Reagent, cyano groups can be introduced on the trimer structure [225,226]. In another study, 2-aminothiophene-3-carbonitrile was covalently linked to the trimer aldehyde in presence of acid to form an azomethine derivative [226]. Furthermore, some cyano derivatives were obtained by treating with t-BuOK the 1, 3- Dithiol-2-one heterocylic ring fused on the central tiophene of the trimer. The resulting compound carries two nitrile moietes linked to the trimer through thioether groups. In a further step, this molecule can be exploited for the preparation of cyclic thioethers [227,228]. Lastly, Hsu et al. synthesized and exploited the electrochemical properties of some fused benzo- and naphtonitrile derivatives [224]. The synthesis of thiophene-EDOT-thiophene trimers with the cyano groups on the side units (3 and 3″ position) was accomplished by a palladium-catalyzed coupling reaction, using organozinc thiophenes carrying a nitrile and dibromothiophene as starting materials [221,223,224]. 

### 3.7. Bromo Groups

Bromine is the most exploited halogen for cross-coupling reactions. Bromo-trimers are normally obtained by bromination of the monomer precursors. In here, the bromination of the thiophene monomers is performed not only in 2 and/or 5 position, that are the ones suitable for the trimer preparation, but also in 3 and/or 4, where other functional groups can be attached by cross-coupling C-C bond formation. Rasmussen et Al. studied extensively how the regioselectivity of bromination is influenced by electronical and sterical effects [229]. Trimers bearing a bromine atom in 3′ were prepared by Stille [230], grignard [149,150], Kumada [215,229,231,232,233,234], Suzuki [154,235,236] and microwaves Suzuki reactions [157]. Dibromo derivates in 3′ and 4′ have been obtained using Grignard [237] organotin [238], Suzuki [239,240,241], Kumada [237] and palladium C-C coupling [156,242]. Finally a 3,3″-dibromoterthiophene was obtained by debromination reaction in 1, 1″ [160] and a 4,4″-dibromoterthiophene was prepared by final formation of central thiophene ring via Lawesson reagent [243]. The synthesis of tetrabromo derivates was achieved by Nighishi coupling [244,245,246,247,248].

The insertion of bromine was also performed on the trimer. By using NBS and AIBN two bromomethyl groups in position 4 and 4″ were attached (Scheme 13, II) [249]. Another way to obtain trimers carrying bromo functional groups is to introduce them as pending group of reactive molecules, for example by reaction of an hydroxyl functionalized trimer with bromoacetyl chloride (Scheme 13, III) [250,251] or with a dibromo alkyl chain [252]. Bromo pending groups were further exploited for the Atom Transfer Radical Polymerization (ATRP) of the trimers [253,254]. EDOT-containing brominated trimer are also reported: 3′-bromoProDOT-like trimer has been obtained using an organotin derivate [155] as well as a 3′-bromo(EDOT-thiophene-EDOT) [152].

### 3.8. Fluoro, Chloro, and Iodo Groups

Few examples of fluorinated substituents are available in literature. A common strategy consist in the substitution reaction between a bromo group attached to the trimer and the desired perfluorinated alcohol molecule, forming an ether. This strategy allowed to connect in 3′ a perfluorinated chain of seven [255] and nine carbons (Scheme 14, I) [249]. Fluorotrimers were obtained by using fluorinated monomers for the C-C cross-coupling reaction. In this way, 3′,4′-difluoroterthiophene (Scheme 14, II) [256], 3′-perfluorohexylterthione [257] and 3,3″-bis(perfluorohexyl)terthiophene [258] were obtained. Also, a difluoro[c]’-fused maleimide [259] and a fused [c]’-perfluorocyclopentane group (Scheme 14, III) were introduced on the trimer [260,261]. An unusual structure containing two terthiophenes bridged by a perfluorocyclopentane ring was employed as photoswitch [262]. There are very few works reporting the synthesis of iodo [263] and chloro [264,265] trimers.

### 3.9. Alcohols

An alcohol pendant group is a chemically useful linker. Trimer functionalization has been achieved by exploiting alcohol reactivity to form several ethers [266,267,268,269,270,271] or esters [253,272,273,274,275]. 3-thiopheneethanol is commercially available and it is the most used precursor for building trimer alcohols, such as [2,2′:5′,2′′-Terthiophene]-3′-ethanol. In alternative, the same structure can be obtained by reduction with lithium aluminum hydride (LiAlH_4_) of Ethyl 2-(2,5-di(thiophen-2-yl)thiophen-3-yl)acetate [275]. The hydroxyl function was also obtained by aldehyde reduction. In this way, methyl alcohol was obtained either in position 3′ [148,276,277], in position 3 [278], or in position 3 and 3″ [279], affording asymmetrical or symmetrical trimers with the alcohol group on the lateral units. An example of asymmetric trimer carrying alcohols on the lateral units is given by the works of the van Esch’s group, where the –OH terminated poly(ethylene glycol) chains are connected in position 3″ and 4 [280,281]. Trimers with central and/or lateral EDOT units were synthesized. Hydroxymethyl EDOT is a commercial molecule and it was inserted both between two thiophene units (Scheme 15, III) [282] and between two EDOT units [271]. Otherwise, central 3-thiopheneethanol was coupled with two EDOT molecules (Scheme 15, IV), forming an EDOT-thiophene-EDOT trimer [271].

### 3.10. Ethers

Ether is one of the most widespread pendant group of terthiophenes. It is known that alkoxy chains increase the solubility and improve the electronic properties of trimers and, therefore, their presence is ubiquitous in the literature. Typically, two O-connected alkyl chains (one per lateral unit) are found in 3 and 3″ [71,283,284,285,286] or in 4 and 4″ position of the trimer (Scheme 16, I) [287,288,289,290,291,292,293,294]. Besides, structures with one [295,296] or two [297,298] alkoxy groups on the central thiophene, in 3′ and 4′ position, were synthesized. Methoxy [283], ethoxy [296], butyloxy [298], pentyloxy [88,287], hexyloxy [290,298], octyloxy [298], decyloxy [290,299] pendant groups, among many, have been reported. Another type of common ether-based pendant group is poly(ethylene glycol). Similarly to the alkoxy groups, poly(ethylene glycol) chains were positioned on both the lateral units (Scheme 16, II) [185,281,286,300] or on the central thiophene [269,301,302,303,304], in order to improve the solubility of the trimer or as linker between the trimer and other materials. Several crown ethers, a special category of poly(ethylene glycol) pendant groups, have been synthesized for cation sensing applications. Different ring sizes were prepared [234,305,306,307,308]. The crown ether was connected directly to the trimer [307,308,309] or by means of other functionalities, such as cyano or styryl lateral groups (Scheme 16, III) [305,306,310,311]. In some works, trimer’s alcohol group has been converted to ether for functionalization purposes [266,267,268,269,270,271]. Finally, EDOT, a very common building unit of trimers, owes its peculiar electronic properties to the cyclic double ether pendant group (Scheme 16, IV).

### 3.11. Thioethers

Several thioethers directly connected to the trimer by sulfur were reported. Alkylsulfanyl chains of different lengths were introduced on the central [228,296,312,313] and/or on the lateral [306,314,315] thiophene units. Additionally, some examples of vicinal sulfurs, in position 3′ and 4′ of the trimer, are reported [227,228,312]. A particular case is the all-sulfur analog of PEDOT, whose effect on the electronic properties of different trimers was scrutinized [227,316]. Fused dithiino spacer groups, with sulfur atoms connected in position 3′ and 4′ of the trimer, were obtained from a 1,3-dithiole-2-thione intermediate [317]. This functionality was exploited for connecting to the trimer some redox-active pending groups, i.e. substituted tetrathiafulvalenes [318,319,320] and fluorenes [318,321]. A similar strategy was covered for the preparation of dithiinoquinoxaline [322] and tetrathianaphtalene [323] units.

### 3.12. Ketones

Ketones are strong electro-withdrawing groups and, if directly connected to the thiophene ring, can increase the electrofilicity of the trimer. Therefore, they are commonly introduced for tuning its electronic properties. Y. Ie et Al. prepared trimer ketones through the insertion of dioxocyclopenta[c]thiophene between two thiophene units by Stille or Suzuki coupling (Scheme 17, I) [259,324,325]. Moreover, the acidic α position between the two ketones was exploited for further functionalizations [324,325]. The benzodithiophenedione structure is another cyclic ketone and serves as acceptor unit in donor-acceptor copolymers for photovoltaic applications (Scheme 17, II). Trimers formed by a central benzodithiophenone linked to two thiophene rings are prepared by Stille coupling and present some variation on the pendant group [242,326,327,328,329,330,331,332,333]. Finally, a terthiophene linked to a non-cyclic β-diketone was also reported (Scheme 17, III) [334].

### 3.13. Aldehydes

The preparation of trimer aldehyde most commonly employs 3-thiophenecarboxaldehyde as a commercially available starting material. This monomer can be inserted as the central unit of the trimer by a bromination step followed by Suzuki coupling with thiophene- or EDOT-boronic acids [276,279,294,335,336,337,338]. The resulting trimer has an aldehyde attached in position 3′. The same structure was also achieved by Grignard metathesis. Nonetheless, in this case it was required the protection of the aldehyde group with 2,2-dimethylpropane-1,3-diol before the coupling reaction [338]. W.-C. Xu et Al. afforded an asymmetric trimer with the aldehyde attached to position 4 of a lateral unit [278]. Trimers with two aldehyde groups were also prepared, in position 3′ and 4′ on the central unit [242], or in position 3 and 3″ on the lateral units [71]. The presence of an aldehyde group connected to the thiophene ring is of great advantage for the functionalization of trimers. For example, several β-substituted trimers were prepared by Wittig’s condensation between the aldehyde and a phosphonium salt [148,279,338,339]. Aldol condensations involving the trimer aldehyde and an amide [340], ketone [341], diketone [337] or malonitrile [294] group were also successfully performed.

### 3.14. Carboxylic Acids

Various trimer carboxylic acids have been prepared, directly connected to the thiophene rings [342,343,344,345] or linked through a spacer. In fact, spacers of one [346,347,348,349,350,351,352], two [279,353,354], three [355], four [354], five [356] or six [350] carbon atoms are reported (Scheme 18, I and II). Benzoic acid, connected in para on position 3′, is also reported (Scheme 18, III) [249,357]. since the Suzuki or Stille coupling are not compatible with the presence of carboxylic acids, this group is normally converted to methyl or ethyl ester before the formation of the trimer [344,346,350,351]. Once the trimer is obtained, the ester is hydrolyzed and the carboxylic acid restored. Therefore, common precursors of carboxylic acid trimers are 3-thiophenecarboxylic acid and 3-Thiopheneacetic acid. Thiophene aldehyde can be converted to α,β-unsaturated carboxylic acid by aldol condensation [279,354]. Furthermore, in order to avoid the esterification and the hydrolysis steps, in some works the carboxylic acid is obtained by hydrolysis of nitrile [313,342,345]. The negative charge of carboxylic acid is widely exploited for promoting attractive interactions between the trimer and other molecules, materials or surfaces [349,350,355,358,359]. Additionally, trimers carrying a carboxylic acid can be easily functionalized by esterification or amidation [343,346,347,360].

### 3.15. Esters

Ester is a multivalent group employed for the preparation and functionalization of trimers. Esters have been introduced for tailoring the physical and electronic properties of trimers and trimer-based polymers [63,242,361,362,363], as intermediates for the preparation of alcohols [272,275,350,364,365] and pyridazinediones [366,367] and as a carboxylic acid protecting groups in the Stille and Suzuki couplings [344,346,350]. The ester formation is a preferential strategy for the functionalization of trimers. The alcohol [250,253,273,274,275,277,364,365,368,369] or the carboxylic acid [302,343,344,346,347] on the trimer is exploited for this purpose (Scheme 19, I and II). In this way, several molecules, nanoparticles and other functional materials were covalently linked to the trimers, including spyropyran [343,370], gold nanoparticles [371], poly(ethylene glycol) [302,372], cellulose [373], methaclyate [277], thiocarbonylthio derivate useful for RAFT (Reversible addition−fragmentation chain-transfer) polymerization [275] and olefin dendrons [374]. Numerous works report the preparation of the 3′,4′ trimer diester (Scheme 19, III) [242,361,362,366,375,376]. Additionally, the work of A. Fazio et Al focuses also on the synthesis of trimers carrying esters on the position 3,4 and 3″,4″ of the lateral thiophenes, rather than on the central one [362].

### 3.16. Amides

The amide bond formation is an excellent approach for the easy and stable attachment of pendant groups to the trimer. Functional pendant groups carrying an amine or a carboxylic acid were attached respectively to trimer carboxylic acid [303,304,377,378,379,380,381] or amine (Scheme 20, I and II) [207,208,209,382]. The amide formation occurs on the side group of the central thiophene either before [377,378,379,380] or after [207,381,382] the formation of the trimer and it is typically achieved by carboxylic acid activation by acyl chloride [207,208,209,378] or a carbodiimide coupling agent [377,379,380,381], followed by amine nucleophilic substitution. Instead, the reaction between a trimer diester and hydrazine leads to the formation of a pyridazinedione pendant group on the central unit (Scheme 20, III) [366,367]. A significant subcategory of trimer amides presents as central unit a thieno[3,4-c]pyrrole-4,6-dione (Scheme 20, IV). This structure is a strong electron acceptor; therefore, it has been widely studied as component of donor-acceptor zwitterionic copolymers. The monomer is prepared by conversion of a 3,4-thiophenedicarboxylic acid [197,202,383] or its cyclic anhydride [383,384,385,386] to acyl chloride and subsequent introduction of the desired amine. Depending on the amine, several linear and branched alkyl chains of different lengths were introduced on the N-position, with the purpose of improving the solubility of the resulting material [383,387,388,389]. The pyrrole-dione group is also found on perylene-fused trimers [185,390,391].

### 3.17. Fused Aromatics

The bandgap of trimers can be tuned efficiently by extending their conjugated system with fused aromatic pending groups. In fact, merging electron donating or electron withdrawing aromatic groups with the thiophene rings leads to changes in the HOMO and LUMO energy of the whole system. Most commonly, the central thiophene is fused with aromatic rings like benzene, pyrazine, thiophene, thiadiazole or imidazole, that can, in turn, be connected or fused with other aliphatic or aromatic groups. All these possibilities leaded to the preparation of a huge variety of terthiophenes with extended aromaticity and new optoelectronic properties. 

Different strategies were developed in order to obtain a trimer with a central benzo[c]thiophene unit between two thiophene units (dithienylbenzo[c]thiophene) (Scheme 21, I). Typically, two equivalents of 2-mercaptopyridine react with 1,2-Benzenedicarbonyl dichloride, forming two reactive thioester bonds [392,393,394,395,396]. Afterwards, the mercaptopyridines are replaced by two thiophene substituents, by addition of 2-thienylmagnesium bromide. Finally, the central thiophene unit is obtained by employment of either phosphorous pentasulfide [397,398], Lawesson’s reagent [392,393,394,395,399] or Davy’s reagent [396]. Benzene-1,2-dicarbaldehyde can be used in the place of 1,2-benzenedicarbonyl dichloride: in this case 2-thienylmagnesium bromide directly reacts with the aldehyde groups, but the oxidation of the resulting alcohols to ketones is then needed before the Lawesson’s reagent step [400,401]. Other similar approaches with slightly different starting materials were successfully performed [402,403,404,405,406,407]. Finally, also the classic Stille or Suzuki coupling is a feasible pathway [390,404,408]. Benzene rings with a variety of substituents were introduced through those methods on the central unit of the trimers, including chloro [394], methoxyl [409], and alkyl groups [393,396], as well as sulfides [393], esters [224], nitriles [224], amides [390,408] and additional aromatic rings [224,405,410]. 

Pyridazine and pyridazine derivatives form a large category of fused aromatic rings on terthiophenes. The thienopyrazine, the central unit of the trimer, acts as electron acceptor, while alkyl-substituted lateral thiophenes act as electron donor. The zwitterionic character of the trimer results in the destabilization of the HOMO and the narrowing of the bandgap. The synthesis of 5,7-di(2-thienyl)thieno[3,4-b]pyrazine (Scheme 21, II) involves a 3′,4′-terthiophene diamine and an α-diketone. The double condensation occurring between amines and ketones forms a pyridazine ring fused with the central thiophene. Therefore, changing the diketone substituents, a large variety of substituted pyrazine rings can be obtained, such as alkyl groups [178,411], unsubstituted and substituted phenyl groups [180,196,412,413,414,415], tiophenes [413,416,417], furans [201,418], pyridines [419,420], phenazine [421], naphtalimide [185], perylene imide [422] carbazole [423], and fullerene [424]. In alternative to α-diketones, an α-diester [167] or an α-diimine [425] can be employed for the obtaining of functionalized thienopyrazine units.

Besides the thienopyrazine, thienothiadiazole is another recurring electron withdrawing thiophene derivative (Scheme 21, III). The formation of the thiadiazole group on the trimer’s central thiophene requires, as for the pyrazine, the presence of a vicinal diamine in position 3′ and 4′. The heterocycle is obtained by reaction with N-thionylaniline [165,173,184,186,189,426,427]. Similarly, the pendant group thienoimidazole (Scheme 21, IV) is obtained from the reaction between a 3′,4′-trimer diamine and trimethylorthoformate [171] acyl chloride or acetic acid in strong acid conditions [428]. Finally, thieno[3,4-b]thiophene is a common monomer employed for the formation of trimers [429,430,431,432]. The free carboxylic acid function can be exploited for the introduction of different chemical modifications [429].

### 3.18. Aryl and Heteroaryl Groups

Aryl and heteroaryl groups have been widely exploited as trimer pendant groups (Scheme 22). The resulting extended aromaticity can improve the optoelectronic properties of the entire conjugated system. Moreover, the interaction between the aromatic pendant group and external molecules or materials have a direct influence on the trimer band gap, changing its electronic response. Therefore, a large number of trimers connected to aryl and heteroaryl groups has been studied. The functionalization of the central thiophene was performed either before [85,249,433,434] or after [232,435,436,437,438] the trimer formation. Common reactions for the formation of the aryl-thiophene bond are the Stille coupling, involving an aryl bromide and an alkylstannyl thiophene [433,436], and the Suzuki coupling, involving an aryl bromide and a thiophene boronic acid [232,434,437,438]. In this way, phenyl groups with different substituents have been introduced both on the central [85,433,435,437,439,440,441] and on the lateral [249,434,441] thiophene units. The reported heteroaryl groups includes pyridines [232,341,436,442,443], oxadiazoles [444,445], triazoles [446], pyrimidines [214,215], and BODIPY [447]. Alkenes and alkynes are frequently employed as connectors between the trimer and an aryl or heteroaryl group. In this way, electron delocalization between the two aromatic systems is preserved. By alkene group formation, benzene [338], naphthalene [448], anthracene [448] and pyrene [449] where connected to the trimer, as well as substituted phenyl groups [195,210,310,338,450,451,452,453], pyridines [148,339,454], thiophenes [454,455,456], and ferrocene [457]. Through alkyne bridge, mono-, bis- and ter-pyridines were attached [162,236,458,459], as well as fullerenes [460,461].

### 3.19. Thiophene S,S-dioxide

The oxidation of the thiophene sulfur affords the corresponding S,S-dioxide and it is a valuable strategy for modyifing the optical and electronic properties of thiophene-based trimers (Scheme 23). The presence of S,S-dioxide allows to tune the HOMO and LUMO energies of the trimers as well as to improve their crystalline organization in semiconducting films. The sulfur oxidation is usually performed with m-chloroperoxybenzoic acid (mCPBA) because of its easy of employment and removal. In alternative, HOF·CH3CN was used as oxygen-transfer agent [464]. Lastly, S,S-thiophene dioxide was obtained from the reaction between thionyl chloride and zirconacyclopentadiene [465]. In order to control the dioxide position in the trimer, it is critical to perform the oxidation on the monomer rather than on the trimer, otherwise all the rings will be oxidized at once. In a C-C cross coupling reaction, the S,S-thiophene dioxide can be found either on the halide or on the organometal reagent. Typically, Stille coupling is employed with this aim [466,467,468,469,470,471]. The preparation of terthiophene 1′,1′-dioxide was also performed using Suzuki coupling: the S,S-dioxide group was formed on a diiodo thiophene and the resulting molecule was further made react with thiophene boronic acid in a Pd-catalyzed reaction [36]. The dioxide has been also introduced on the EDOT unit of terthiophenes [36,469,472]. 

### 3.20. Metal Complexes

Several pendant groups have been employed in order to introduce metal binding sites on the trimers. Depending on the choice of these groups, different metals were targeted for the formation of organometallic complexes. For example, diphenylphosphine was employed for binding noble metals like gold [473], iridium [474], osmium [475], palladium [476,477], rhodium [266] and ruthenium [475,478]. In alternative, gold and platinum were binded on alkyne groups located at position 3 and 3″ of the trimer lateral units [161,479]. Also aromatic rigid amines are a class of useful ligands: a bipyridine-functionalized trimer was employed for binding cobalt [458], while terpyridines for binding ruthenium [459,462,480,481] and porphyrins for cobalt [482], copper [482], nickel [483] and zinc [482,484]. Similarly, cadmium [485], copper [101,311,486], nickel [311] and zinc [101] were coordinated by the two nitrogen and two oxygen atoms of the ligand N,N’-bis(salicylidene)-3,4-diaminothiophene, located on the central position of the trimer. A comparable strategy was chosen for the coordination of uranium [487]. bisterthiophene complexes were achieved by the coordination of two terthiophene dithiolenes with gold, nickel and palladium [488]. Erbium complexes, instead, were obtained by linking the erbium atom to two acetate groups located in position 3′ and 4′ of the trimer. Also organomolybdenum complexes were prepared by employing functionalized terthiophenes [489,490]. Some examples are shown in Scheme 24.

### 3.21. Charged Trimers

Charged trimers show enhanced water solubility and therefore are very interest-ing for bioapplications. The positive charge can be conferred by a phosphonium salt (Scheme 25, I) [148,491] or a nitrogen atom, in the form of either a viologen moiety (Scheme 25, II) [492] or an ammonium salt [271,493]. Negatively-charged trimers have the advantage to be self-doped, in a similar fashion with respect to the PEDOT:PSS system. The negative charge is conferred by a sulfonate residue, linked to a terthio-phene molecule [494] or to an EDOT-thiophene-EDOT trimer (Scheme 25, III). In this case, the sulfonate is connected to the trimer through an ether spacer [271,495,496,497,498].

### 3.22. Crosslinked Trimers

As small class of terthiophene-bases structures which is worth to mention is the one where a trimer is linked to another, resulting in a possible crosslinking agent for further polymerization or for 3D networks. Even if not crowded, this group of trimers spaces across a vast range of applications: from optic to energy conversion, including band-gap studies and synthesis of dendrimers. Among these crosslinked trimers, the simplests consist of two terthiophenes directly linked one to each other [499] or connected by a saturated alkylic moiety [250,500]. Binaphthol [270] and fluorinated spacers are also reported [501], as well as several substituted phenyl spacers [437,439,487,502,503,504]. Dendrimer structures have been also prepared [268,269,272]. Some examples are shown in Scheme 26.

## 4. Bioapplications

The unique properties of terthiophenes and derivatives, including ease of functionalization and good biocompatibility, make these materials excellent candidates for several biomedical and biological applications. In recent years, their use and diffusion has quickly increased in multiple fields. In this part of the review, a summary of biological and medical studies involving terthiophene and derivatives is presented. For clarity purposes, this section is formed by two subsections: in the first one, the applications regarding the direct use of terthiophene and terthiophene derivatives are described, whereas, the second section is dedicated to the applications of the polymers based on these structures (Figure 1).

### 4.1. Terthiophene and Terthiophene Derivatives

A large number of published studies reports the use of terthiophene and its derivatives as photosensitizers (PS) [505,506,507]. Commonly, PS work transferring the photon energy to oxygen molecules, to generate reactive oxygen species (ROS), such as singlet oxygen (^1^O_2_), peroxide (O_2_^2−^), superoxide (O_2_^•−^) and hydroxyl radicals (OH^•^), to react with macromolecules, such as protein, lipid or DNA, and lead to oxidative damage under the irradiation of light with appropriate wavelengths.

#### 4.1.1. Pesticides

Today, PS based on terthiophenes are useful alternatives to traditional pesticides for population control of insect pests. For example, some derivatives showed photo-induced inhibitory and cytotoxic effects against *Spodoptera litura*, otherwise known as the tobacco cutworm or cotton leafworm [508]. Stability and insecticidal activity of the trimer were studied for the first time by the group of Zhang. The results assessed the terthiophene high toxicity against *Aedes albopictus* (tiger mosquito) and *Plutella xylostella* with a maximum absorbency at 0.480 after UV irradiation for 200 min [509]. In another study, it was founded that the photo-cytotoxicity of terthiophene on the ovarian Tn5B1-4 and Sf-21 cells can be enhanced by increasing concentration and irradiation time [510]. Similarly, Zhang, tested the ROS formation ability of terthiophene against *Aedes aegypti* larvae, leading to cell death patterns. In here, the authors have studied the ROS effect on mitochondria as function of the terthiophene concentration. The exposure to low trimer concentrations induced apoptosis, while moderate concentrations promoted autophagy through induction of ROS, inhibited apoptosis, and induced necrosis. In contrast, high concentrations of trimer induced high levels of ROS, which, causing mitochondrial dysfunction, directly induced cell necrosis [511]. Overall, the hope is that photoactive compounds can act as efficient insecticides while reducing their environmental adverse effects. Huang et al. evaluated the exposure risks on human 293 cells and insect Tn-5B1-4 cells to photo-activated trimers at different doses. Photo-activated trimers exhibited dose-dependant toxicity on the growth of 293 cells (EC_50_ = 6.23 μg/mL) and Tn-5B1-4 cells (EC_50_ = 3.36 μg/mL). Therefore, the photoactivated terthiophene might be a potential factor in human mutagenic progression. Nevertheless, additional studies are necessary in this regard to clarify the toxicity mechanism of photoactivated trimers, as well as to evaluate and minimize their environmental risks and bioaccumulation [512]. Besides their photo-activated toxicity, terthiophenes have shown toxic effects even in absence of light. In nature, some plants secrete specific thiophene derivatives, including terthiophenes, in order to inhibit the growth of undesired plagues or neighbors [513]. The herbicidal activity of terthiophene was studied by Dong et Al. on *Digitaria sanguinalis*, *Arabidopsis thaliana* and *Chlamydomonas reinhardtii*. The authors isolated the terthiophene from *Flaveria bidentis* (L.) *Kuntze* and they studied how it affected the plant protein expression by dimensional gel electrophoresis and liquid chromatography tandem mass spectrometry. It was observed a decrease in the expression of proteins related to energy production and carbon metabolism and it was suggested that this effect was due to the interaction of terthiophene with a plant transketolase [514]. Other non-photo-induced toxicity effects of terthiophene were reported against the Formosoan subterranean termite [515], the mosquitoes Aedes aegypti [516], and the cyst-forming nematode in Zea mays [517].

#### 4.1.2. Antifungal and Antibacterial Activity

Terthiophene and its derivatives are good candidates for photodynamic therapy (PDT) thanks to their ability to generate ROS under ultraviolet (UV) and visible (Vis) light. For example, Sortino et al. exploited these structures for treating oropharyngeal candidiasis, a common fungal infection in immunocompromised patients. The authors investigated the toxicity of terthiophene under UV light irradiation against the Candida species responsible for the disease. Two variables were taken into account: exposure irradiation time and distance to the irradiation source. The optimal condition for the in vitro antifungal activity of terthiophene was found to be 5 min of UV light irradiation from a distance of about 6 cm. Surprisingly, terthiophene was able to kill twenty different resistant strains of Candida at the low concentration of 0.31 g/L [505]. Similarly, the same group exploited the plant of *Porophyllum obscurum* as a source of new PS with potential use in PDT of oropharyngeal candidiasis cases. The antifungal photosensitive activity of different extracts from *Porophyllum obscurum* was evaluated by using microdilution and bioautographic assay under UV irradiation. This work established a correlation between the composition of the different thiophene extracts with respect to their photo-induced antifungal effect [506].

Liu and coworkers tested the antibacterial activity of the marigold extract containing terthiophene in common food. The authors observed an inhibitory effect on the growth of *E. coli*, *Salmonella* and strains of *Penicillium*: the antibacterial effect of the terthiophene extract increased by increasing the concentration and the exposition time [518].

#### 4.1.3. Sensing

Thanks to their large aromatic conjugated structure, it is not surprising that terthiophenes derivatives have found large application as fluorescent dyes. In Figure 2a four examples of 5-R-terthiophene derivatives designed for this purpose are shown. For example, Chow et Al. demonstrated the ability of 5-(4-ethynyl-N,N-dimethylaniline)-terthiophene (Tt1 in Figure 3a) to enter HeLA cells and the easy detection of its fluorescence emission (Figure 2b), thus providing a new strategy for the design of molecules for two-photon imaging in vitro [519]. Furthermore, cytotoxicity assays revealed the low reduction of viable cultured HeLa cells after 24h in presence of the trimer, thus confirming the high biocompatibility of this compound.

The terthiophene derivatives depicted in Figure 3a have been successfully exploited for the sensing of toxic molecules. Guo and co-workers synthesized a terthiophene functionalized with 1,3-indendione (3TI, Figure 2a) or barbituric acid (3TD in Figure 2a) for the rapid and highly sensitive colorimetric and fluorimetric detection of cyanide (CN-) in real water samples and farm products, as well as for bioimaging in living cells [520,521]. Similarly, the authors synthesized a dithiane derivate (3TH, Figure 2a) for colorimetric detection and ratiometric fluorescence enhancement response to Hg2+ [522]. According to the authors, all the above-mentioned terthiophene-based sensors present multiples advantages, including high fluorescence brightness, fast response time (30 s), minimal pH dependence in the physiologically relevant pH range and excellent selectivity in presence of other competitive ions. Furthermore, given their high water solubility, good biocompatibility and low cytotoxicity, subcellular imaging of the target ions in living cells was achieved with low detection limits, i.e. 31.3 nM 22.6 nM for CN- with 3TI and 3TD respectively, and 62 nM for the detection of Hg(II) with 3TH [520,521,522].

Mono- and bi-functionalized terthiophene-based trimers are useful structures for the development of chemiresistors. In fact, the high polarizability of sulfur electrons may provide a variety of intra- and intermolecular interactions, which can improve charge transport. In addition, the chemical and physical properties of these structure can be easily tuned through convenient structural modifications. In this respect, Liu and co-workers demonstrated that the terthiophene functionalization with aromatic structures is an effective way to enhance their photochemical and thermal stability, thus overcoming some limitations for their sensing application. The authors prepared two chemiresistive sensing devices by direct deposition of two different naphthalene-functionalized trimers (NA-3T and NA-3T-NA) on glass substrates (Figure 3), and demonstrated their efficiency in the detection of gaseous biogenic amines (BAs), important contaminants mostly found in the spoiling process of foods [523]. In particular, the NA-3T-NA-based sensor showed higher sensitivity for trimethylamine (TMA), with an experimental detection limit of 22 ppm. The difference in the sensing performances between the two chemiresistive sensors was ascribed to the different packing of the terthiophene derivatives into the prepared films [523].

In another work, a sensor was obtained by using terthiophene derivatives as molecular linkers in order to tether the antibodies to magnetic nanoparticles. In fact, the direct binding of biomolecules to the nanoparticle would have been more complicated because of the steric hindrance. Instead, thanks to a thin self-assembled monolayer of terthiophene, linkers on the nanoparticles allowed a denser functionalization and subsequently improved the sensitivity of the device. This strategy was successfully covered for the detection of progesterone (LOD = 0.013 ng·ml^−1^) by using an ultrasensitive surface plasmon resonance (SPR) gold chip, modified as described in Figure 4 [304]. In another example, COOH-biphenyl-functionalized trimers self-assembled on gold nanoparticles were used to covalently immobilize brain-derived neurotrophic factor antibodies (BDNF) through the amide bond formation between the amine groups of antibody and the carboxylic acid groups of the trimers [524]. The resulting microfluidic immunosensor was employed to detect the release of BDNF from various cancer cells by the effect of various exogenous activators (ethanol, K^+^, and nicotine), proving the potential of this sensing system in drug screening and therapeutics.

Finally, a folic acid sensor was prepared by electropolymerizing a bis-terthiophene-based dendron onto a quartz crystal microbalance (QCM) [503]. The sensor response showed good linearity within the concentration range of 0–100 μM of folic acid, with a detection limit of 15.4 μM and good selectivity against other competitive molecules.

#### 4.1.4. Pharmacological Activity

It is known that terthiophenes exhibit various biological effects, such as photo-activated cytotoxicity[525] and anti-bacterial activity [505,509]. However, the anti-cancer activity of terthiophenes in human cells and its molecular mechanism are still poorly understood. Jian and coworkers evaluated the antiproliferative activity of terthiophenes against several human tumor cells, i.e. K562, MCF-7, A549, and HCT116, revealing that these structures promote the cell apoptosis [526]. Jin and coworkers reported the photo-activated toxicity of terthiophene extracted from *Echinops grijsii Hance* roots on HepG2, K562, HL60 and MCF-7 human tumor cell lines, but in the in vivo animal experiment it was not observed any significant anti-tumor activity [527]. Jang et al. evaluated by MTT assay the cytotoxicity against human ovarian cancer cells (SKOV3) of a series of terthiophenes isolated from *Eclipta prostrata L*. These compounds showed a significant cytotoxicity, with IC_50_ values ranging from 24.57 to 58.20 μM. The group suggested terthiophene-methanol as potential candidate for additional studies in order to evaluate its potential as an anti-cancer agent [528].

Terthiophene showed antitumor effects on several cancer cell lines, including ovarian cancer cells [525,526,527]. The discovery of the anti-proliferative effect of thiophene derivatives by the group of Preya dates back to 2007. In here, it was investigated the molecular mechanism behind the anti-proliferative effect of terthiophene-methanol. It was found that this structure is able to inhibit the growth of human ovarian cancer cells by arresting the S phase of the cell cycle via induction of ROS stress and DNA damage. Thus, it was prepared a potential agent for the treatment of ovarian cancer based on terthiophene-methanol (Figure 5) [525].

Saito et al. founded that local oxidation reactions on the cell membrane produce submicron-sized holes. Remarkably, after the perforation the cells were still viable. Therefore, they designed and fabricated a rod-shaped device in which the membrane perforation function was made possible by the presence of terthiophene as PS. The cell membrane perforation was successfully achieved with a light intensity of 0.82 W/cm^2^ for 30 s. The authors demonstrated that the cells, impermeable to a certain fluorescent dye before perforation, could instead uptake it once treated in the way described above. However, the group of Saito suggested that in the near future the UV-active PS should be replaced with a visible- or IR-active PS in order to reduce the risk of mutagenesis induced by UV irradiation [529].

### 4.2. Polimerized Trimers

#### 4.2.1. Sensing

The detection of proteins, vitamins, and other nutrients in food and biological samples is a task of high biological, technological, and pharmaceutical importance. Therefore, it is necessary to develop rapid, selective, and sensitive methods for their reliable determination, which are capable of satisfying the high product monitoring standards demanded by the government regulations. For this purpose, terthiophene-based conducting polymers have proven to be an excellent choice, because of their simple functionalization and relatively good stability in air. Consequently, polyterthiophenes have been employed for electrode surface modifications, allowing the development of biosensing devices for multiple purposes. For example, polyterthiophenes (pTTP) are the first compounds used as electronic transducers for protein recognition. Amine or carboxylic acid pending groups are typically exploited for the attachment of biomolecules.

##### Poly(amino-terthiophene)s

Amine groups are commonly used for the conjugation of small biomolecules, such as vitamins or drugs that can further interact with the target protein. For instance, biotin, a type B vitamin with a carboxylic acid group, was covalently immobilized onto a conducting polyterthiophene coated electrode by exploiting the primary amine side groups of the polymer. The as-prepared sensor was succesfully employed for the detection of avidin, a highly stable glycoprotein found in egg-whites, showing a linear response between 4 × 10^−14^ and 3 × 10^−4^ mol/L and a detection limit of ca. 10^−14^ mol/L [210,211]. Due to its large-scale usage as antioxidant, ascorbic acid is another highly interesting target molecule. Abdelwahab et al. developed an aminopyrimidyl-functionalized-pTTP based sensor for ascorbic acid showing excellent selectivity between 10 to 200 mM and a detection limit of 1.4 mM [212]. Moreover, the same polymer was used for the detection of rocuronium (Figure 6), a residual drug that may cause serious safety issues to the patients. This molecule is a neuromuscular blocking agent used in anesthesia that facilitates tracheal intubation by muscle relaxation. Aminopyrimidyl-functionalized-pTTP was covalently binded to phosphatidylinositol lipid via amine group, obtaining an efficient sensor for monitoring rocuronium in blood samples without pre-treatment, with a dynamic range between 0.025 to 10 ug/mL and a detection limit of 3.83 ng/mL (Figure 6) [216]. A sensor for the simoultaneous detection of piroxicam, an anti-arthritis drug, and its major interferences (L-ascorbic acid, tyrosine, and uric acid) in urine samples was also developed [530]. In this case, the aminopyrimidyl groups of the pTTP backbone were linked to graphene oxide, which directly interacts with the target molecules. Recently an heterocyclic derivate has been used to detect paracetamol with a minimum detection limit of 60 nM [531].

##### Poly(acid-terthiophene)s

Carboxylic or boronic acid pTTP derivatives have been covalently functionalized with antibodies, that allow the selective recognition of targeted biomolecules. Following this strategy, disposable electrodes for amperometric detection of specific biomarkers were developed. These sensors (see Figure 7 for schematic examples) showed high sensitivity towards cardiac troponin I, glycated hemoglobin and glucose and glutathione disulfide, thus high potential for the diagnosis of acute myocardial infarction, diabetes and brain disorders respectively [532,533,534,535]. Following a similar protocol was studied a glucose sensor based on pTTP coated with AuZn oxide layer [536]. In another study, the fabrication of an ultrasensitive electrochemical immunosensor for detecting human immunoglobulin G (IgG) [537]. In here, the target protein IgG was sandwiched between the anti-IgG antibody, covalently attached to the pTTP via amide bond, and the Ag (I)-cysteamine complex (Ag–Cys) adsorbed onto gold nanoparticles (AuNPs)–anti-IgG. The detection signal is originated from the electrochemical stripping of Ag from the adsorbed Ag–Cys complex on the AuNPs–anti-IgG. This sensor showed a wide dynamic range with a detection limit of 0.4 fg/mL.

Immunosensors based on benzoic acid-pTTP (BA-pTTP) showed high efficiency for the in vitro monitoring of inducible nitric oxide synthase (i-NOS), a family of enzymes catalyzing the production of nitric oxide. The detection of i-NOS is as an indirect measurement of endocrine disrupters, which downregulates i-NOS and produce adverse developmental, reproductive, neurological, and immune effects in both humans and wildlife [538,539]. The sensor, which showed a limit of detection of 0.2 ng/mL, was prepared by self-assembly of electropolymerized pTTP on gold nanoparticles (AuNP) followed by electropolymerization on a glassy carbon electrode surface (Figure 7b). 

Furthermore, the benzoic acid functionality showed excellent sensitivity for the electrochemical detection of toxic products released by normal or diseased cells. In the work of Kim et Al., the detection of nitric oxide (NO) produced from cancer cells (LOD = 7.7 × 10^−9^ M) was achieved by using ZnO nanoparticles immobilized on a BA-pTTP/rGO composite layer [540]. In another study, silver nanoparticles (AgNPs) attached to a BA-pTTP/carbon nanotubes composite was tested as a biosensor for the detection of H_2_O_2_ in urine samples showing a fast response time (below 5s) and a LOD of 0.24 mM [541]. Lastly, phthalate was analytically monitored with a microfluidic device coupled to a BA-pTTP electrochemical biosensor, with the aim to evaluate the effect of endocrine disruptors. The high uptake ability of the biosensor towards soluble phthalate esters in aqueous media (LOD = 12.5 pM) was achieved by controlling the surface charge and hydrophobicity through assembling with a lipid and a cationic molecule attached to the BA-pTTP matrix [542].

#### 4.2.2. Antibacterial Activity

The group of Rodrigues described the preparation of an antiwetting and self-cleaning superhydrophobic pTTP film and its effect on enabling or inhibiting the adhesion of proteins and bacterial cells on its surface (Figure 8). The authors could tune the polythiophene wettability by simply changing its redox state via potential switching. For instance, the undoped pTTP film, which is superhydrophobic, inhibits the adhesion of fibrinogen proteins and *E. coli* cells. On the other hand, the doped film, which is hydrophilic, leads to increased attachment of both proteins and bacteria. Overall, manipulating the wettability affect the adhesion of fibrinogen and *E. coli* [543].

#### 4.2.3. Tissue Engineering

As many polythiophenes, polyterthiophenes are considered attractive materials for their implementation in synthetic cellular scaffolds, as they are biocompatible and can closely interact with cells and tissues both in an electrical and biological way. For example, Quigley and co-workers tested an alkoxy-functionalized pTTP together with two different dopants as substrates for the growth and differentiation of primary myoblasts [544]. The authors found that *p*-toluenesulphonate–doped polymers were significantly smoother and hydrophilic than the perchlorate counterpart. Such properties and the presence of methoxy groups had a significant effect on the primary myoblasts attached onto the polymer surface and resulted in the promotion of different specific cellular responses, e.g., proliferation vs. differentiation, in absence of other biological agents. In another work, the carboxylic acid moiety of pTTP was exploited for the grafting of fibronectin-derived Arg-Gly-Asp (RGD) peptides. Through this strategy, an electronically conductive and biocompatible surface was obtained. Afterwards, this material was successfully employed for the the attachment and growth of human dermal fibroblasts, proving its potential as bioactive scaffold for tissue engineering applications [345].

#### 4.2.4. Pharmacological Activity

Polyterthiophene has been used as platform for the controlled release of dexamethasone, a synthetic glucocorticoid anti-inflammatory drug. The controlled release profiles were established using a range of electrochemical stimulation protocols over a period of 24 h. Interestingly, the redox state of the polyterthiophene was found to be critical for the controlled release of the dexamethasone. In fact, in the reduced state the amount of dexamethasone released from the polyterthiophene under the electrostimulation protocols was at therapeutically relevant levels, with a maximum release of ≈80 g/cm^2^. On the contrary, in the oxidized state the rate of release of dexamethasone was significantly impeded with ≈40 g/cm^2^ released over 24 h [545]. 

Prion diseases are neurodegenerative infectious disorders characterized by the deposition of β-sheet-rich aggregates. The infectious agent is termed prion and Aguzzi and coworkers reported the use of polythiophenes for the inhibition of its propagation by stabilizing the prion proteins. They tested on cerebellar organotypic cultured slices and on mouse prion proteins in vitro a wide variety of polythiophenes, including polyterhtiophenes, in their anionic, cationic or zwitterionic form, using the enzyme-linked immunosorbent assay (ELISA). Overall, the polythiophenes reduced the infectivity of prion-containing brain homogenates and cerebellar organotypic cultured slices and decreased the amount of scrapie isoform of prion proteins. Nevertheless, the antiprion activity of these compounds cannot be attributed to the charge type of their side chains, because anionic, cationic, and zwitterionic compounds reduced prion infectivity to a similar level. Thus, the activity of these polythiophenes appears to be an intrinsic property of the polymer backbone itself. Finally, some of the polythiophenes synthesized displayed the ability to cross the blood-brain barrier and therefore may represent promising candidates for further in vivo studies [546].

## 5. Future Perspectives

The synthetic strategies, the structural variety, and the biological applications of terthiophenes were discussed in this review. It is clear how the ease of synthesis and the variation and modification of these structures has made possible their implementation into a wide palette of new advanced technologies, consecrating them as the following generation of thiophene-based functional materials.

Nevertheless, although the chemical and optical properties of terthiophene derivatives have been extensively investigated, their applicative potential still needs to be fully explored. In particular, the employment of these structures in biology and medical science studies is rather limited compared to their widespread use for optoelectronics and energy conversion applications. The reason could lie in the sometimes-challenging communication across the fields: on one side, the unawareness of the synthetic tools available, on the other, their unacknowledged value for developing useful technologies, which hampers the progress of transversal research. This review attempted to overcome some of these difficulties, providing with simple words and clear examples the complementary information to promote interdisciplinarity. Many possible bioapplications of trimer-based materials were discussed in this review, including photodynamic therapy and plagues treatment, as well as the development of platforms to diagnose and treat a wide variety of diseases. In particular, the successful use of terthiophene as an organic PS has been abundantly reported. Nonetheless, these PS are activated by UV light, whose exposition is potentially mutagenic, and more effort should be done to make them active under the safer visible or infrared wavelengths.

In summary, the enormous amount of approaches and possibilities available for the synthesis, functionalization, and application of thiophene-based trimers is reflected in the increasing number of outstanding scientific results. Our belief is that the day in which these materials will have a positive impact on everybody’s life is now close. With this work, we hope to have provided a useful help to whomever may want to participate in this goal.

## Data Availability

Not applicable.

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
