# Peer review of "Thiophene-Based Trimers and Their Bioapplications: An Overview"

_polymers, 2021, doi:10.3390/polym13121977_

Round 1

Reviewer 1 Report

This paper reviews synthesis and bioapplications of terthiophene derivatives. The manuscript features several synthetic methodologies and recent activities on sensing, antibacterial, and pharmacological activity. The paper covers many recent papers, and it is properly and well organized. The information would be useful for readers of this journal. Therefore, the reviewer recommends this paper for publication in Polymers

Minor point:

The reviewer is curious about that chemical structure in Scheme 21 III) and Scheme 24 II) would be matched with the references (189 and 481). 

Author Response

Reviewer 1:

This paper reviews synthesis and bioapplications of terthiophene derivatives. The manuscript features several synthetic methodologies and recent activities on sensing, antibacterial, and pharmacological activity. The paper covers many recent papers, and it is properly and well organized. The information would be useful for readers of this journal. Therefore, the reviewer recommends this paper for publication in Polymers.

Minor point:

The reviewer is curious about that chemical structure in Scheme 21 III) and Scheme 24 II) would be matched with the references (189 and 481).

We thank the reviewer for the nice work done. We have corrected the errors, in the case of the 189 was a paper by the same author and in the case of 481 was changed with the following reference.

We have really appreciated the patient of the reviewer to control and carefully read all the manuscript.

Reviewer 2 Report

In this manuscript, the authors systematically reviewed the synthetic strategies, structural variety and biological applications of terthiophenes. They have pointed the easiness of synthesis, variation and modification of these structures, which made them potential in some new advanced technologies. Meanwhile, they also have shown the rough edges of their applicative potential, especially the employment of these structures in biology and medical science studies. At last, they expressed their good perspective that these materials will have a positive impact on everybody’s life. The manuscript may easily bring readers to the research area of erthiophenes and their derivatives. We support the publication of the manuscript in this journal, while we expect the authors can address several problems as follow:

  1. The authors should carefully revise some schemes including the font, type size, line spacing and overall arrangement so that they can present the corresponding meanings more intuitively and aesthetically. Such as Scheme 4, 5, 7, 8, 12, 16 and so on.
  2. Some format problem should be revised, such as the position of the serial numbers of the references should be consistent, all before or after (Line 158, 190….) the punctuation; the format of Pd(Cl)2 and AlCl3.
  3. Where is the section 5 before Future Perspectives?
  4. References related to the bioapplications after 2019 have not been listed, we think the latest research progress should be more attractive to the readers.

Author Response

Reviewer 2:

In this manuscript, the authors systematically reviewed the synthetic strategies, structural variety and biological applications of terthiophenes. They have pointed the easiness of synthesis, variation and modification of these structures, which made them potential in some new advanced technologies. Meanwhile, they also have shown the rough edges of their applicative potential, especially the employment of these structures in biology and medical science studies. At last, they expressed their good perspective that these materials will have a positive impact on everybody’s life. The manuscript may easily bring readers to the research area of erthiophenes and their derivatives. We support the publication of the manuscript in this journal, while we expect the authors can address several problems as follow:

    The authors should carefully revise some schemes including the font, type size, line spacing and overall arrangement so that they can present the corresponding meanings more intuitively and aesthetically. Such as Scheme 4, 5, 7, 8, 12, 16 and so on.

We have strongly appreciated the comments of the reviewer, we have adjusted as our best all the molecular size in all the schemes.

    Some format problem should be revised, such as the position of the serial numbers of the references should be consistent, all before or after (Line 158, 190….) the punctuation; the format of Pd(Cl)2 and AlCl3.

Subscript and superscript have been corrected as remarked by the reviewer, as well as the formatting typos.

    Where is the section 5 before Future Perspectives?

We are very sorry about this silly mistake; it has been corrected.

    References related to the bioapplications after 2019 have not been listed, we think the latest research progress should be more attractive to the readers.

In the last year, only few papers have been published relatively to our topic and less regarding/matching the describe applications: as kindly suggested by the reviewer the literature have been updated and the relative comments added in the text.

We thank a lot the reviewer for the comments and the input to improve the manuscript.